# Proliferation and Immune Response Gene Signatures Associated with Clinical Outcome to Immunotherapy and Targeted Therapy in Metastatic Cutaneous Malignant Melanoma

**DOI:** 10.3390/cancers14153587

**Published:** 2022-07-22

**Authors:** Fernanda Costa Svedman, Ishani Das, Rainer Tuominen, Eva Darai Ramqvist, Veronica Höiom, Suzanne Egyhazi Brage

**Affiliations:** 1Unit for Head and Neck, Lung, and Skin Cancer, Karolinska University Hospital, 171 76 Stockholm, Sweden; fernanda.costa-svedman@regionstockholm.se; 2Department of Oncology-Pathology, Karolinska Institutet, 171 64 Solna, Sweden; ishanidas09@gmail.com (I.D.); rainer.tuominen24@gmail.com (R.T.); veronica.hoiom@ki.se (V.H.); 3Unit for Cytologi and Pathology, Karolinska University Hospital, 171 76 Stockholm, Sweden; eva.darai-ramqvist@regionstockholm.se

**Keywords:** melanoma, targeted therapy, immunotherapy, gene expression profiling, biomarker

## Abstract

**Simple Summary:**

The introduction of treatment with targeted therapies and immunotherapies has dramatically changed the outcome for patients with advanced cutaneous melanoma. However, only a subset of the patients has durable benefits from the treatment. This exploratory study aims to identify genes/gene signatures as predictive biomarkers for treatment outcomes in melanoma. Targeted transcriptomics and gene set enrichment analysis (GSEA) were applied in 28 melanoma samples collected before receiving treatment. Thirteen patients were treated with targeted therapy (TT) and 15 patients were treated with immune checkpoint inhibitors (ICI). Up-regulation of genes involved in immune processes was associated with a better outcome of TT. Down-regulation of proliferation and up-regulation of allograft rejection gene sets favored ICI patients. Further follow-up of the inverse relation between proliferation and allograft rejection gene signatures and relation to outcome is warranted.

**Abstract:**

Targeted therapy (TT), together with immune checkpoint inhibitors (ICI), has significantly improved clinical outcomes for patients with advanced cutaneous malignant melanoma (CMM) during the last decade. However, the magnitude and the duration of response vary considerably. There is still a paucity of predictive biomarkers to identify patients who benefit most from treatment. To address this, we performed targeted transcriptomics of CMM tumors to identify biomarkers associated with clinical outcomes. Pre-treatment tumor samples from 28 patients with advanced CMM receiving TT (*n* = 13) or ICI (*n* = 15) were included in the study. Targeted RNA sequencing was performed using Ion AmpliSeq ™, followed by gene set enrichment analysis (GSEA) using MSigDB’s Hallmark Gene Set Collection to identify gene expression signatures correlating with treatment outcome. The GSEA demonstrated that up-regulation of allograft rejection genes, together with down-regulation of E2F and MYC targets as well as G2M checkpoint genes, significantly correlated with longer progression-free survival on ICI while IFNγ and inflammatory response genes were associated with a better clinical outcome on TT. In conclusion, we identify novel genes and their expression signatures as potential predictive biomarkers for TT and ICI in patients with metastatic CMM, paving the way for clinical use following larger validation studies.

## 1. Introduction

Treating metastatic cutaneous malignant melanoma (CMM) patients with targeted therapy (TT) and immune checkpoint inhibitors (ICI) has significantly improved progression-free survival (PFS) and overall survival (OS), as demonstrated in multiple phase III clinical trials during the last decade [1,2,3,4,5,6,7,8,9]. The use of BRAF inhibitors (BRAFi) (vemurafenib, dabrafenib, encorafenib) in combination with MEK inhibitors (MEKi) (cobimetinib, trametinib, binimetinib) has proven to perform better than chemotherapy and BRAFi in the 50% of CMM patients with *BRAF V600* mutations [3,5]. The median PFS with the combination is around 1 year compared to approximately 7 months with BRAFi alone and 2 months with chemotherapy [3,4,5]. The median OS has improved from less than 1 year with chemotherapy to around 2 years with BRAFi and MEKi in combination, reaching 33 months with encorafenib and binimetinib [1,2,3,4,5,10].

Of the ICI, ipilimumab, a cytotoxic T-lymphocyte–associated antigen 4 (CTLA4) inhibitor, was the first to be approved, with an increase in 5 year-OS rates from 8% to 18% compared to dacarbazine, which was the standard of care approximately 10 years ago [11]. Ipilimumab was followed by other ICIs, nivolumab and pembrolizumab, that inhibit programmed cell death 1 (PD-1). Treatment with nivolumab has improved PFS from 2 to 5.5 months and the OS from 11 to 37 months compared to dacarbazine [7,8,12]. Pembrolizumab has improved PFS from 3.4 to 8.4 months and the median OS from 15.9 to 32.7 months compared to ipilimumab [13,14].

Further improvement in outcome in this patient population has been demonstrated with the combination of nivolumab and ipilimumab, which outperformed monotherapy of either agent with a PFS of 11 months and a 5-year OS of 52% compared to 44% with nivolumab and 26% with ipilimumab [9,15]. Nivolumab and pembrolizumab are better tolerated than ipilimumab, with rates of grade 3 and 4 adverse events around 15% compared to around 25%, respectively. The rates of grade 3–4 adverse events are further increased to 55% with the combination treatment using current scheduling and dosing [7,8].

Even though the breakthrough in CMM treatment with TT and ICI has been impressive, most of the patients exhibit primary (although not frequent in TT) or acquired resistance. There are clinical factors known to be associated with poorer treatment outcomes to these treatments. High lactate dehydrogenase (LDH) baseline levels, more than two organs compromised by metastasis, and the sum of lesions diameter (SLD) of more than 65 mm predict the worse effect of TT. Impaired performance status, presence of liver metastasis, and symptomatic brain metastasis are also examples of baseline clinical features correlating with worse outcomes in ICI-treated patients [16]. However, more than half of the patients without these clinical parameters will develop the progressive disease within 3 years after treatment starts [17].

For TTs, multiple mechanisms of resistance such as *NRAS* mutations, *BRAF* amplification, *MEK* mutations, *NF1* mutations, *AKT* amplification, loss of *PTEN*, and overexpression of HGF and receptor tyrosine kinases (RTKs) like MET, PDGFRβ, and IGF1R have been described [18,19]. For ICI, examples of tumor characteristics that have been associated with decreased treatment sensitivity include low tumor mutational burden (TMB) (whole-exome sequencing) and low T cell-inflamed gene expression profiling GEP (RNA sequencing). The GEP included 18 inflammatory genes related to antigen presentation, chemokine expression, cytolytic activity, and adaptive immune resistance [20]. In addition, previous transcriptome analyses in melanoma samples showed that upregulation of immune checkpoint genes and enrichment of genes involved in PD-1 signaling and IFN-γ signaling (among other immune processes) in gene set enrichment analysis (GSEA) was more pronounced in patients with response to ICI [21,22].

Although extensive research in the biomarker field has been performed, there are still no predictive tests with sufficient evidence to support clinical treatment decisions for CMM beyond *BRAF* mutation analysis that predicts response to TT. The identification and validation of further predictive biomarkers are an unmet need as these treatments are associated with high costs and, in some cases also, substantial toxicity.

In this exploratory study, we have performed gene expression profiling of tumor samples taken from patients with metastatic CMM before receiving treatment with TT (*n* = 13) or ICI (*n* = 15), with the aim to identify novel biomarker candidates associated with treatment outcome. A fine needle or core biopsy was taken from an accessible metastasis, making it possible to follow both the effect of treatment on the same metastasis as well as the overall clinical response.

## 2. Material and Methods

### 2.1. Patients

Fine needle aspiration (FNA) or core tumor biopsies from 28 patients with metastatic CMM were collected and immediately put in RNA later (to stabilize and protect cellular RNA) at Karolinska University Hospital, Stockholm, Sweden, between March 2012 and March 2017. Thirteen of these patients were treated with TT and 15 with ICI. The tumor samples (FNA or core biopsy) were taken before treatment started with either TT or ICI. Biopsy material taken on progression from the same metastatic lesion as the baseline biopsy was available from two patients. The study was approved by the Stockholm Regional Ethics Committee, Karolinska Institutet, and was performed according to Good Clinical Practice/the Declaration of Helsinki. All included patients have signed the informed consent form before study entry. Clinical information of the study population such as gender, age, metastatic classification (M) at study start (following the American Joint Committee on Cancer 7th edition) [23], *BRAF* mutation status, details about type, line and duration of treatment, and LDH pre-treatment levels is presented in Table 1 and in Appendix A. All 13 CMM patients in the TT cohort and 14 of 15 CMM patients in the ICI cohort were treatment naïve. One subject in the ICI group was treated with 2 prior chemotherapy regimens (temozolomide and carboplatin/paclitaxel) before treatment with ipilimumab (data available in Table 1, Appendix A).

The biopsies were taken from melanoma metastases localized in the skin, subcutaneous tissue, and lymph nodes in all patients, with the exception of one subject with a liver biopsy (see detailed information in Appendix A). Most of the patients were treated outside of a clinical trial, except for three patients in the ICI cohort who were treated with pembrolizumab in the Keynote-252/ECHO-301 study [24]. Our standard follow-up routines for patients treated outside of clinical trials were followed, which include medical visits every 4–8 weeks and radiological evaluation with computerized tomography (CT), magnetic resonance imaging (MRI), or positron emission tomography (PET) scan every 8–12 weeks.

The response to treatment of the patients included in the Keynote-252/ECHO-301 study was evaluated according to the protocol specifications, whereas for the other patients, it was based on joint evaluation of clinical/radiological investigations evaluated by radiologists and oncologists in connection with case discussions during our weekly tumor boards. Overall clinical response and tumor-specific response (of the biopsied metastasis) were evaluated separately. Patients were categorized as having disease control (DC) if they achieved complete response, partial response, stable disease, or as non-responders (NR) if patients had progressive disease according to our clinical/radiological evaluation described above.

The PFS was calculated from the date of treatment start until confirmed disease progression by imaging or until the date of death or last follow-up, whichever came first. Detailed information on the respective cohorts about the response to treatment and PFS are available in Table 1 and in Appendix A and a flow chart of the study design is shown in Figure 1A.

### 2.2. RNA Extraction

RNAlater was removed from FNA and core biopsies, followed by RNA extraction using the AllPrep DNA/RNA/miRNA kit according to the manufacturer´s protocol (Qiagen, Hilden, Germany). RNA quantity and quality measurements were performed using Agilent Bioanalyzer 2100 instrument (Agilent Technologies Inc., Santa Clara, CA, USA). The RNA was stored at −80 °C until analysis.

### 2.3. Targeted Sequencing Using Ion AmpliSeq™

To identify genes associated with response to treatment (both overall clinical response and biopsied tumor-specific response) and PFS, targeted sequencing of 20,802 different transcripts was performed using the Ion AmpliSeq ™ transcriptome human panel (Thermo Fisher Scientific, Waltham, MA, USA) that recognizes >95% of all RefSeq genes with one amplicon designed for each gene. RNA was used as input material and amplicons were sequenced using the Ion Proton systems from Life Technologies (Carlsbad, CA, USA) as a service at the Uppsala Genome Center, Uppsala University, Sweden.

BAM files were imported into the Genomics Suite^®^ 7.17.1222 software from Partek^®^ (St. Louis, MO, USA) and analyzed using their built-in gene expression workflow. Briefly, for each sample, the total number of alignments, total number of reads, and percentage of reads that overlap completely, partially, or not with exonic regions, were determined. Number of counts for each transcript was normalized using the reads per kilobase per million reads (RPKM) method. A comparison of mRNA abundance of candidate transcripts among samples was made using the RPKM values.

The RPKM data was then exported into the Qlucore Omics Explorer 3.6 software from Qlucore (Lund, Sweden) for the bioinformatic analysis. Standard score (z-score) normalization calculation was performed and a fixed fold-change of 1 was used for the two-group comparison. T-test, linear regression, and rank regression analyses, adjusted for multiple testing using the Benjamini–Hochberg method, was used to identify differentially expressed genes between DC and NR to either treatment and gene expression levels relationship to PFS. The cut-off used for inclusion of genes in the analysis was >10 RPKM in at least 50% of the samples filtering low expressed genes to avoid noise and decrease the number of statistical tests and the batch effect was eliminated. A gene set enrichment analysis (GSEA) with 50 hallmark gene sets from the Molecular Signature Database (MSigDB) was performed, which includes all genes with the same cut-off parameters as above [25].

### 2.4. Immunohistochemistry

Paraffin-embedded tissues obtained from CMM patients were deparaffinized in xylene (3 × 5 min each) followed by rehydration in EtOH 99%, (3 × 2 min each), EtOH 95% (2 × 2 min each), EtOH 70% (1 × 2 min) and finally rinsed in tap water. Citrate buffer (pH 6.0, Sigma Aldrich, St. Louis, MO, USA) was used for antigen retrieval. Samples were then cooled, rinsed in tap water, and blocked for endogenous peroxides using H_2_O_2_ (3 mL of 10% H_2_O_2_ (Sigma Aldrich) in 60 mL deionized water) for 30 min, followed by rinsing in tap water and 1 × TBS. Tissue sections were then circled with a hydrophobic pen and blocked with 1% BSA (prepared in 1 × TBS) for 30min at room temperature (RT). Antibodies were diluted in 1% BSA and sections were incubated overnight with primary ORC1 antibodies (Abcam, Cambridge, UK) at 4 °C. The following day, the sections were washed in 1 × TBS (3 × 5 min each), followed by incubation with anti-rabbit (Vector, BA-1000), diluted to 1:200 in 1 × TBST (0.02% Tween-20), and incubated at RT for 30 min. The sections were washed in 1 × TBS (3 × 5 min each). Following this, the sections were incubated at RT for 30min with ABC kit solution (Vector labs, Newark, CA, USA) according to the manufacturer’s protocol, followed by washes in 1 × TBS (3 × 5 min each). All sections were stained with DAB for 10 min and counterstained with haematoxylin. Sections were then dehydrated in EtOH 70% (1 × 2 min), EtOH 95% (2 × 2 min each), EtOH 99%, (2 × 2 min each), xylene (2 × 5 min each). Slides were thereafter mounted using Pertex (Histolab Products AB, Stockholm, Sweden) and imaged using an Olympus Provis microscope (Tokyo, Japan). Independent evaluation of all slides was performed by three observers (F.C.S, I.D, S.E.B). In case of discrepancies between observers, a consensus was reached on further review. The intensity (negative, low, moderate, or strong) and proportion of ORC1 positive tumor cells were evaluated and specimens with strong nuclear staining in >20% of the tumor cells were regarded as high expression.

### 2.5. Statistical Analysis

Clinical variables (incl gender, age, stage, and LDH levels) have been correlated with overall clinical response to the different treatments using Fisher’s exact test, unpaired *t*-test, or logistic regression. For survival analysis with PFS as the time variable, Cox regression was used for clinical variables (univariate and multivariate analysis). The correlation between PFS and gene counts for TT was analyzed using linear regression (since all had progressed at the study cut-off) and for ICI therapy, the rank regression was used. An unpaired *t*-test was used to detect differentially expressed genes (DC vs. NR). Survival curves were estimated using the Kaplan–Meier method and compared statistically using the Log-rank (Mantel-Cox) test with the GraphPad Prism 9.0.0 software (GraphPad Software Inc., San Diego, CA, USA).

## 3. Results

### Elevated Baseline LDH Levels Correlates with Shorter PFS in Patients Treated with Targeted Therapy

We have correlated baseline clinical characteristics (age, gender, LDH, stage (AJCC 7th edition [23]) with overall clinical response and PFS in patients treated with TT and ICI. The median PFS for patients with DC and NR was 245 days and 60 days, respectively, in the TT group and 1260 and 55 days in the ICI group (Figure 1B,C). None of the clinical characteristics significantly correlated with overall response neither to TT nor to ICI. For PFS as an endpoint, LDH showed a significant association with PFS in the TT group (HR 1.12, 95% CI 1.00–1.25, *p* = 0.03), while in the ICI group, it did not reach significance (HR 1.33, 95% CI 0.98–1.79, *p* = 0.067). The association between elevated baseline LDH and shorter PFS on TT remained significant in the multivariate analysis adjusted for gender, age and stage (HR 1.12, 95% CI 1.00–1.27, *p* = 0.04), while it remained borderline significant for ICI (HR.1.42, 95% CI 0.99–2.05, *p* = 0.058).

As an FNA or a core biopsy was taken from the tumor for the transcriptomic analysis prior to treatment, we were able to assess associations of both systemic efficacy as well as the effect of the specific lesion from which a biopsy had been taken (tumor-specific). Eleven out of 13 CMM patients were classified as having DC on TT considering the tumor-specific response, whereas 7 out of 13 patients showed an overall clinical response. The tumor-specific response and overall clinical response were thus not consistent in four patients (Appendix A). To find transcriptomic signatures associated with tumor-specific response and overall clinical response, transcriptomic analysis was performed on tumors from patients who had DC or were NR to TT. Gene set enrichment analysis (GSEA) using MSigDB´s Hallmark Gene Set Collection showed no significant signatures (Appendix A).

The 50 differentially expressed genes between DC and NR (tumor-specific and overall clinical response) with the lowest significant *p*-values are shown in a hierarchical clustering heatmap (Figure 1D,E; Appendix A). Twenty-three of the genes associated with tumor-specific response had a false discovery rate <20%, including *IGF,* whereas only one gene associated with clinical response, *PRKG2*, had FDR <20% (Figure 1F). *PRKG2* has been demonstrated to bind to and inhibit the activation of receptor tyrosine kinases [26,27].

When correlating transcriptomic data to PFS in GSEA, significantly enriched gene signatures were observed in pathways, including inflammatory and IFN-γ responses with FDR < 5% and IL6-JAK-STAT3 signaling with FDR < 10% (Figure 1G, Appendix A). Five of the 50 genes with the lowest significant *p*-value had FDR < 20% (Figure 2A, Appendix A). All these five genes, *PSMB8*, *STAT1*, *GCH1*, *GBP1P*, and *CD8A*, are known to be induced by IFN-γ (*IFNG*). *PSMB8*, downstream of *STAT1* in the IFN-γ signaling pathway, significantly correlated with *STAT1* expression (r = 0.97, *p* < 0.0001). High expression of *PSMB8* was associated with longer PFS (Figure 2B).

In two of the 13 patients, biopsy material taken on progression from the same metastatic lesion as the baseline biopsy was available. In both cases, there was a reduction in the IFN-γ response (NES = −2.05; q = 0.063) as well as in the interferon-alpha (IFN-α) response (NES = −1.77; q = 0.10) at progression compared to before treatment.

Ten of 15 patients were classified as having tumor-specific DC, four were NR, and one patient was not evaluable for tumor-specific responses to ICI therapy. Twelve patients were classified as having DC, whereas 3 were NR for overall clinical response (tumor-specific response and overall clinical response were not consistent in one patient—Appendix A).

GSEA was also performed here to identify transcriptomic signatures differentiating tumor-specific and overall clinical DC and NR to ICI therapy from each other. The GSEA for the ICI cohort showed that the downregulation of genes in proliferative processes, including E2F targets, G2M checkpoint, and mitotic spindle, correlated with the response (Figure 2C, Appendix A) but FDR < 20% was only observed for a tumor-specific response. In addition, MYC targeting genes were also significantly upregulated in NR compared to DC, supporting the proliferative signature of NR. Although not significant, an upregulation of IFN-γ response genes in DC was observed (Figure 2D, Appendix A). Furthermore, an inverse correlation between proliferative gene sets and allograft rejection and IFN-γ response gene sets was demonstrated (Figure 2C,E).

The 50 differentially expressed genes with the lowest significant *p*-values between DC and NR to ICI therapy, tumor-specific and overall clinical response are shown in hierarchical clustering heatmaps (Figure 3A,B and Appendix A). Six of the genes in the tumor-specific response list and seven in the clinical response list had FDR < 20%. Several cell cycle-related genes were found in the top 50 lists, including *ORC1*, *MCM7*, and *CKS2*.

For the GSEA of PFS, we divided the patients into three groups (<6 months, >6 months and <2 years, and >2 years) and performed rank regression. We found that enrichment of decreased expression of gene sets involved in proliferation (E2F targets, MYC targets, and G2M checkpoint) and increased expression of genes in allograft rejection (immune process) were significantly correlated with longer PFS (FDR < 20%) (Figure 3C, Appendix A). When comparing these with enriched gene signatures (FDR < 20%) in the TT cohort, we found an overlap regarding allograft rejection genes (Figure 3D). Additionally, a hierarchical clustering heatmap illustrates the top 50 significant genes correlating with PFS to ICI (Figure 3E, Appendix A); however, none of these genes had FDR < 20%. Six genes overlapped in the three top 50 lists for tumor response, overall clinical response, and PFS, including *MAGOH*, *C16orf87*, *ORC1*, *NAA35*, *PWP2*, and *SIAH2*. Total DGE lists are provided in Appendix A.

*ORC1* was one of the 6 common genes expressed correlating with both tumor-specific response, overall clinical response, and PFS in the ICI Ampliseq cohort (Figure 4A). We have thus assessed ORC1 protein expression by IHC in 28 melanoma tumors from a small cohort of 24 patients (Appendix A) treated with ICI, of which 6 patients overlapped with the Ampliseq cohort. These tumors were collected up to 8 months before treatment started (surgery or biopsy). ORC1 expression was mainly observed in the nucleus (Figure 4B). When multiple tumors were analyzed from the same patient, the one with the highest ORC1 expression was chosen for correlation with response. Response data was not available in one patient due to death shortly after initiating therapy, but 4 of 7 NR (57.1%) and 3 out of 16 (18.8%) with DC had strong nuclear staining in >20% of the tumor cells (*p* = 0.14). High nuclear expression of ORC1 was associated with shorter PFS (*p* < 0.0001; Figure 4C). These findings are in accordance with the mRNA data.

## 4. Discussion

We identified gene expression signatures in pre-treatment melanoma metastases associated with clinical outcomes following TT or ICI therapy by performing GSEA and applying the MSigDB´s Hallmark Gene Set Collection. We found that up-regulation of gene sets involved in immune processes (IFN-α, IFN-γ, and inflammatory responses as well as IL6-JAK-STAT3 signaling) were associated with longer PFS on TT and observed a reduction in the IFN-α and IFN-γ response gene sets in biopsies taken at progression. Moreover, decreased expression of proliferation gene sets (E2F targets, G2M checkpoint, MYC targets, mitotic spindle) and increased expression in the allograft rejection gene sets correlated with response and longer PFS on ICI therapy. Although not statistically significant, an increased enrichment of IFN-γ response genes was also observed in patients with DC and longer PFS in the ICI group, indicating an agreement with the literature [22].

Our immune gene signatures associated with prolonged PFS on TT are in accordance with Wongchenko et al.´s gene expressing profiling of 223 *BRAF* mutated melanoma tumors from patients treated with BRAFi (vemurafenib) alone in two clinical trials (BRIM-2 and BRIM-3) where an immune signature significantly correlated with a longer PFS [28]. The immune signature (including *CD8A,* which was also detected in our cohort) was still relevant for PFS in an independent validation cohort, including 99 samples from patients treated with vemurafenib plus the MEKi cobimetinib (within the coBRIM trial). This signature was also correlated with known prognostic clinical baseline characteristics (LDH level, performance status, liver metastasis, and tumor burden) and shown to be an independent factor for longer PFS to TT in the group with normal LDH [29]. High expression of genes in the IFN-γ signature was also found to be a prognostic factor in *BRAF* mutated melanoma patients with resected stage III A–C (AJCC 7th edition) disease regardless of whether they received dabrafenib and trametinib or placebo in the adjuvant setting [30].

The *CD8A* gene expression mirrors CD8+ T-cell infiltration and a decrease in CD8+ T-cell numbers was observed in melanomas at progression on TT compared to at the baseline levels. [31]. Our results showing an increase in the alpha subunit of CD8 (CD8A) being significantly associated with longer PFS in the TT patient group can be an indication of the importance of tumor-directed immunity in TT (Appendix A). The expression of *PSMB8* (another top candidate gene for TT), known to be increased by IFN-γ and downstream of STAT1, has been shown to promote T cell infiltration in the tumor microenvironment by enhancing the repertoire of antigen presentation and to be an independent predictive biomarker for durable responses to ICI therapy (adjusted for tumor mutational load, IFN-γ signature, PDL-1 expression, and T-cell infiltration) [32]. These mechanisms may also explain our findings where high expression of *PSMB8* and *CD8A* predicts better outcomes for TT, indicating that *PSMB8* and T cell infiltration could potentially be treatment-independent biomarkers. High expression of the *PSMB8* gene was not statistically significantly correlated with a better outcome of ICI in our cohort, which could be explained by the small number of patients. However, cross-resistance between TT and ICI is observed in daily clinical practice and has also been also reported in the literature [33]. Randomized clinical trials are therefore investigating the optimal sequencing of TT and ICI to avoid the development of cross-resistance.

Our findings in the ICI cohort with an inverse correlation between proliferative and immune signatures are in line with a recent publication by Grasso and colleagues [22]. They found in GSEA of RNA sequenced melanoma samples from 84 patients (included in CheckMate 038 trial) that high baseline and on treatment expression of IFN-γ signature correlated with response to ICI. In addition, patients responding to ICI also had lower expression of E2F targets, G2M checkpoint, MYC targets, and mitotic spindle [22]. A proliferation signature has also been associated with poorer outcomes in a BRAFi alone (vemurafenib) patient cohort; however, it did not have a sustained relevance in patients treated with vemurafenib plus MEKi (cobimetinib) [28]. High proliferation does appear to have a stronger association with resistance to ICI than to TT since BRAFi monotherapy is no longer a relevant therapy regimen. However, it has been demonstrated that blocking the cell cycle with a CDK4-inhibitor can overcome BRAFi resistance [34].

*ORC1*, one of the genes in the proliferation signature, was found to be overexpressed in non-responders in our ICI cohort and associated with shorter PFS. *ORC1* encodes for one of 6 ORC key proteins responsible for initiating DNA replication. It binds with high affinity with the DNA at the origin of replication sites, assembling with other proteins (CDC6, CDT1, and MCM) to form pre-replicative (PR) complexes triggering the DNA replication process [35,36]. ORC1 has been shown to be an unfavorable factor in melanoma in agreement with our findings (ORC1. Pathology Atlas. The Human Protein Atlas. Available at: https://www.proteinatlas.org/ENSG00000085840-ORC1/pathology (accessed on 17 December 2020).

However, there is no prior data demonstrating that ORC1 could have a predictive role in therapy with ICI in melanoma. In addition, *MCM7* in the PR complex was also associated with worse clinical outcomes to ICI therapy in our cohort. MCM7 has been suggested as an unfavorable factor in other tumor types and significantly correlated with Ki67 expression [37,38]. However, MCM7 may be a more sensitive marker than Ki67, as previously demonstrated in lung cancer [37]. Importantly, ORC (and other PR proteins) should be further considered in the diagnosis and treatment of melanoma and other tumor types [39].

In the context of our data indicating that high proliferation is associated with poorer outcomes following ICI therapy, a significant improvement in PFS and OS by combining ICI with antiproliferative drugs has been demonstrated for other indications, i.e., chemotherapy plus ICI in non-small cell lung cancer [40], tyrosine kinase inhibitors plus ICI in renal cell cancers [41], and chemotherapy plus ICI in triple-negative breast cancer [42] are noteworthy. Similar combination strategies should be investigated in metastatic CMM patients with high expression of proliferative genes and low expression of immune genes, given that these patients seem to respond worse to ICI.

An interesting observation in our cohort is that tumor-specific response and overall clinical response were not consistent in 5 out of 28 patients highlighting the importance of taking tumor heterogeneity into consideration when studying biomarkers in melanoma [43]. Furthermore, our data was generated in a small retrospective cohort without any control group and the results must therefore be interpreted cautiously, but the results are highly consistent with previously published larger studies.

## 5. Conclusions

Our gene expression profiling identifies multiple genes with a strong association with outcome and confirms reported gene signatures associated with clinical outcomes in tumors from CMM patients receiving TT or ICI therapy. Taken together, our results show that enrichment of gene sets involved in proliferation predicts the worse outcome for patients, whereas enrichment of immune genes is favorable for response and PFS. These findings support the inclusion of RNA sequencing in clinical melanoma studies combining ICI with antiproliferative agents to assess if these gene signatures can predict which patients may have benefited from combination therapies.

## Figures and Tables

**Figure 1 cancers-14-03587-f001:**
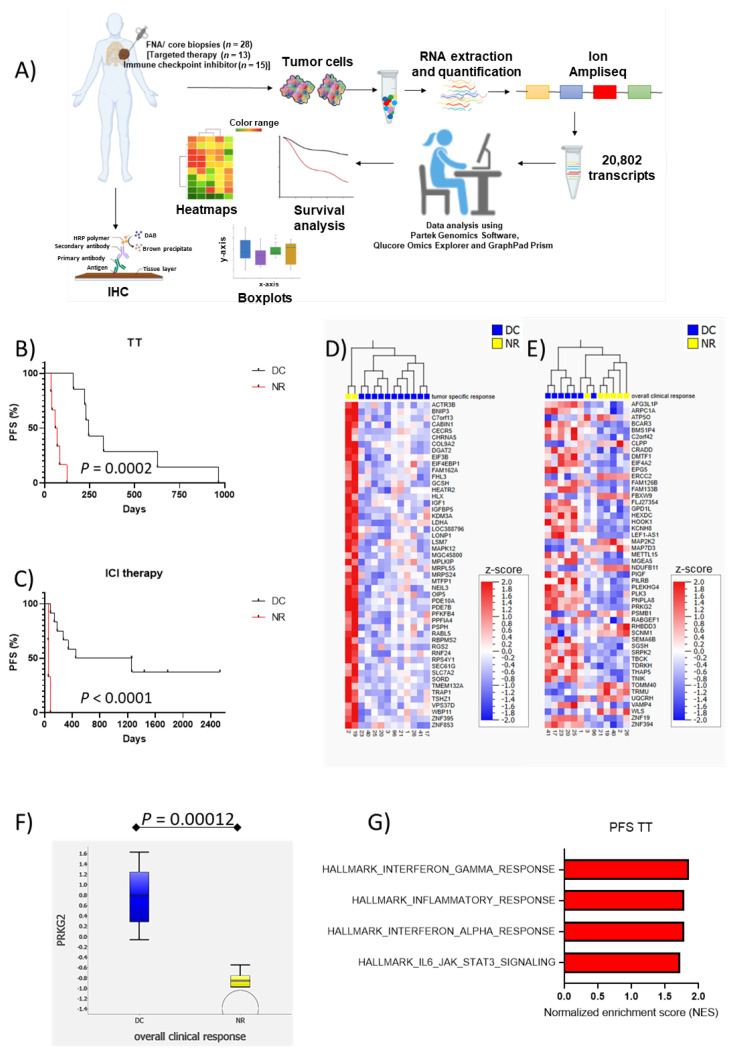
Inflammatory and IFN-***γ*** response signatures are associated with PFS in patients treated with targeted therapy (TT). (**A**) A flow chart of the study design. Fine needle aspirate (FNA) or core biopsy samples and clinical data were collected from 28 Stage IV cutaneous malignant melanoma (CMM) patients before treatment with TT or immune checkpoint inhibitor (ICI) therapy. RNA from tumor samples were analyzed by targeted sequencing using Ion AmpliSeq™. This data has then been used to assess the relationship between RNA expression levels and patient outcome, progression-free survival, and treatment response (tumor-specific and overall clinical) in patients treated with either TT or ICI. Finally, one of the candidates was validated using immunohistochemistry on an independent cohort. The flow chart was created with BioRender.com (last accessed on 7 July 2022). (**B**,**C**) Kaplan–Meier curves demonstrate progression free survival (PFS) for patients with disease control (DC) and non-responder (NR) to TT and ICI, respectively. (**D**) Heatmap for hierarchal clustering of the top 50 differentially expressed genes between tumor-specific DC and NR to TT. (**E**) Heatmap for hierarchal clustering of the top 50 differentially expressed genes between overall clinical DC and NR to TT. (**F**) Box plot showing expression of *PRKG2* in patients with DC and NR to TT (Tukey’s range *t*-test). (**G**): Gene set enrichment analysis (GSEA) showed significantly enriched immune gene sets in relation to PFS after TT (FDR < 10%).

**Figure 2 cancers-14-03587-f002:**
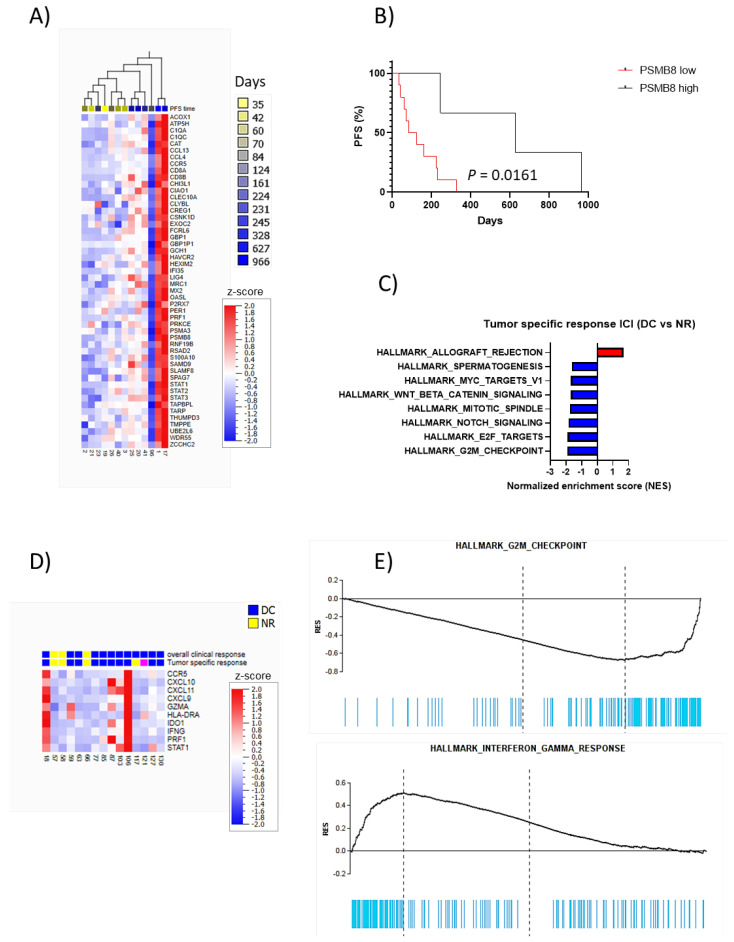
High expression of *PSMB8* is associated with longer PFS in patients treated with TT (**A**): Heatmap for hierarchal clustering of the top 50 differentially expressed genes in relation to PFS in patients treated with TT. (**B**): Kaplan–Meier curves showing effects on PFS for patients treated with TT based on the high or low expression of *PSMB8* (cut-off: 3rd quartile). (**C**): GSEA shows gene sets with the highest normalized enrichment score when comparing tumor-specific DC versus NR after ICI treatment. All except allograft rejection have an FDR < 20%, which has an FDR of 23%. (**D**): Heatmap performed for expression of genes in the IFN-γ signature, previously reported in the literature, in relation to the tumor and clinical response to ICI treatment in our cohort; DC (blue), NR (yellow), and not evaluable (pink). (**E**): GSEA plots showing a reverse expression between G2M checkpoint genes and IFN-γ response genes, with a relative downregulation of G2M checkpoint genes and upregulation of IFN-γ response genes in DC in relation to NR in patients treated with ICI.

**Figure 3 cancers-14-03587-f003:**
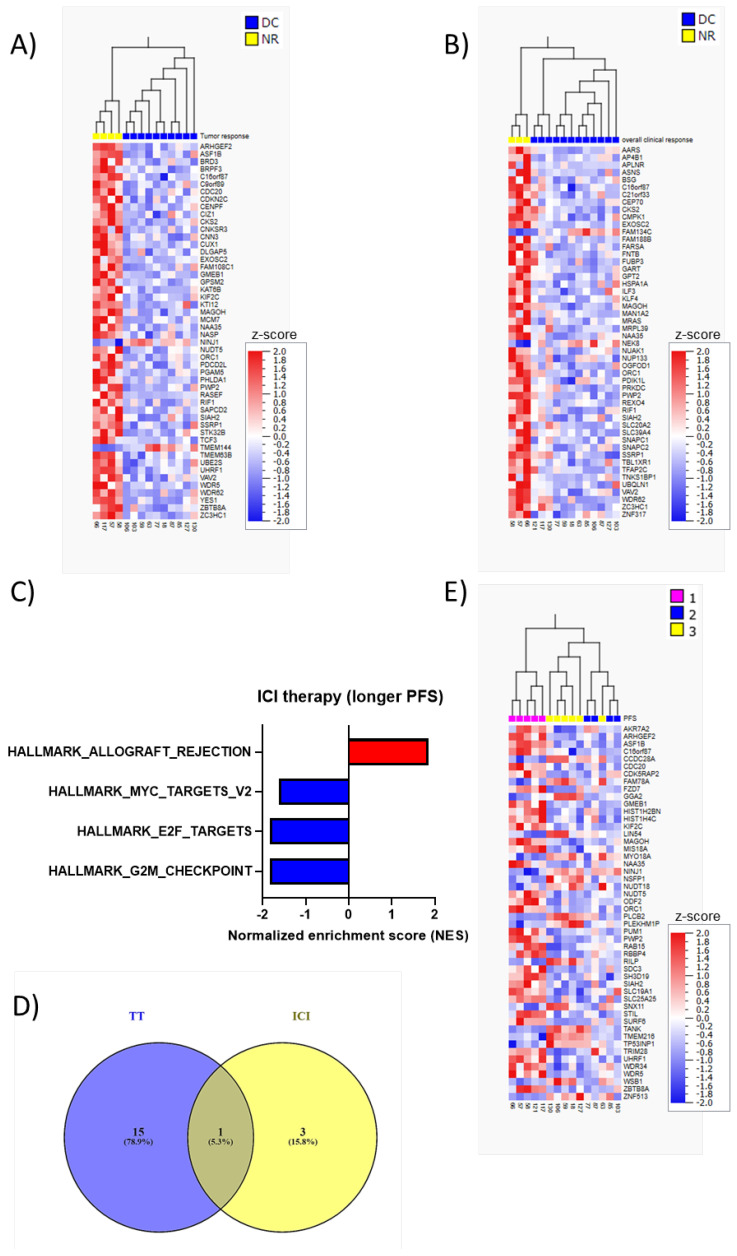
Low proliferative and high allograft rejection signatures are associated with longer PFS in patients treated with ICI. (**A**,**B**): Heatmaps for hierarchal clustering of the top 50 differentially expressed genes between DC and NR (tumor-specific and overall response, respectively) to ICI treatment. (**C**): GSEA shows enriched gene sets in relation to PFS when divided into 3 groups: PFS < 6 months, PFS > 6 months and ≤ 2 years, and PFS > 2 years PFS after ICI therapy. (**D**): Venn diagram showing the overlap between enriched gene sets (FDR < 20%) in relation to PFS in the TT and ICI cohort. (**E**): Heatmap for hierarchal clustering of the top 50 differentially expressed genes in relation to PFS to ICI treatment when divided into three groups (PFS < 6 months, PFS between 6 months and 2 years, and PFS > 2 years).

**Figure 4 cancers-14-03587-f004:**
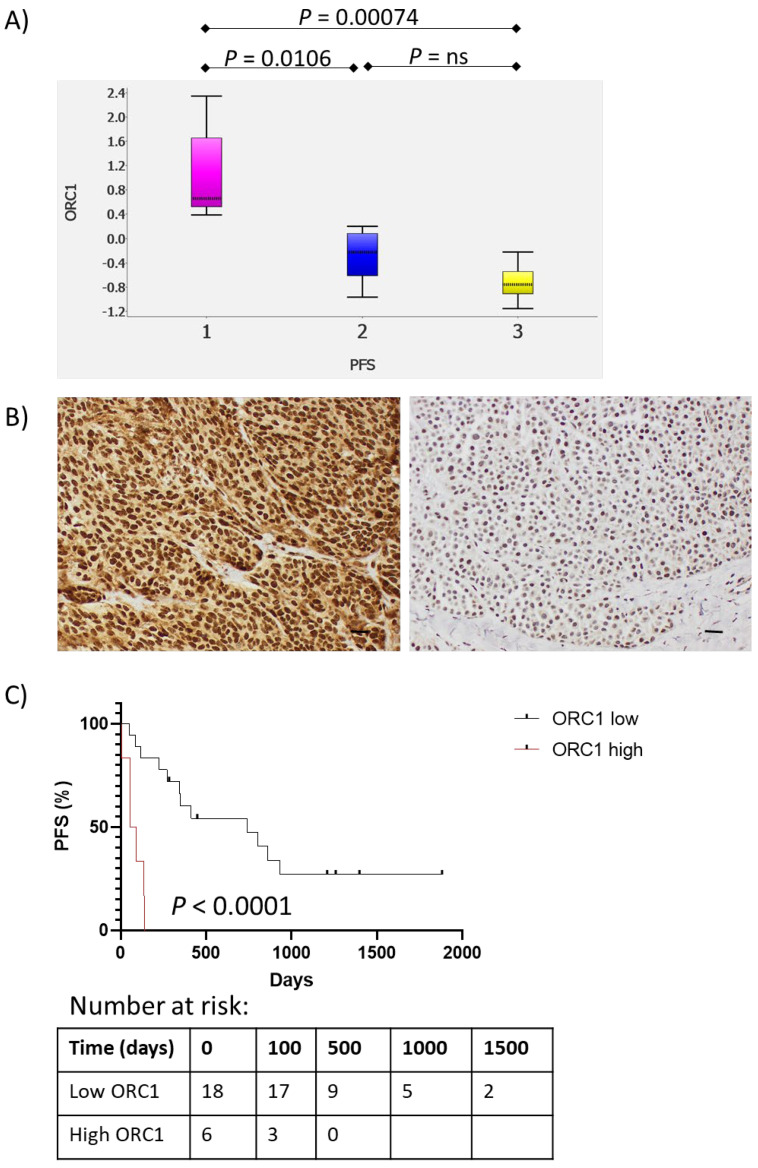
High ORC1 protein expression correlates to a poor clinical outcome. (**A**) Box plots showing a correlation between *ORC1* gene expression and PFS after ICI treatment where patients were divided into three PFS groups: <6 months (pink), between 6 months and 2 years (blue), and >2 years (yellow) (Tukey´s range *t*-test). ns = not significant (**B**) Images of high protein expression of ORC 1 (left) by immunohistochemistry (IHC) in one case with short PFS and ORC1 low expression (right) in a patient with long PFS after ICI therapy. (**C**) Kaplan–Meier curves showing effects on PFS after ICI treatment based on high or low protein expression of ORC1 (IHC). Scale bar = 250 µm.

**Table 1 cancers-14-03587-t001:** Baseline clinical characteristics of patients receiving targeted therapy (TT) or immune checkpoint inhibitor (ICI).

Variable	TT (*n* = 13)	ICI (*n* = 15)
Gender		
Male	9	9
Female	4	6
Age (years old)		
Median (range)	61 (42–86)	73 (49–84)
M1 ^1^ stage		
M1a	1	2
M1b	1	4
M1c	11	9
LDH (microKat/L) ^2^		
Median (range)	5.3 (2.6–30.1)	4.1 (2.6–14.2) ^3^
BRAF status		
Mutation	13	4
Wild type	0	11
Therapy		
BRAFi	10	0
BRAFi+MEKi	3	0
anti-PD-1 ^4^	0	14
anti-CTLA-4	0	1
Line of treatment		
1st	13	14
2nd	0	0
3rd	0	1 ^5^
Response (biopsied tumor)		
Disease control	11	10
Non-responder	2	4
Not evaluable	0	1 ^6^
PFS		
Median (range), days	161 (35–966)	343 (52–2533) ^7^

^1^ According to the American Joint Committee on Cancer Melanoma Staging 7th edition. ^2^ Normal value < 3.5 microkat/L. ^3^ Pre-treatment LDH level not available from 1 patient due to hemolysis. ^4^ Three patients were treated within the clinical trial NCT02752074. Two of them received pembrolizumab plus epacadostat. ^5^ Patient treated with 2 lines of chemotherapy (temozolomide and carboplatin+paclitaxel) before ipilimumab. ^6^ Not evaluable due to the tumor was operated before evaluation of treatment effect. ^7^ Five patients were still responding at study cut-off on 31 August 2020 and one missed follow-up on 30 April 2019. LDH = lactate dehydrogenase; PFS = progression free survival.

## Data Availability

Data is available upon request.

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
