# Peer review of "Proliferation and Immune Response Gene Signatures Associated with Clinical Outcome to Immunotherapy and Targeted Therapy in Metastatic Cutaneous Malignant Melanoma"

_cancers, 2022, doi:10.3390/cancers14153587_

Round 1

Reviewer 1 Report

N/A

The paper entitled “Proliferation and immune response gene signatures associated with clinical outcome to immunotherapy and targeted therapy in metastatic cutaneous malignant melanoma” by Svedman et al. is an interesting paper. The authors used targeted sequencing of advanced melanoma biopsy tissues before targeted therapy or immunotherapy to identify gene signatures responsive to drug treatment. Although the sample size is limited, the discovered gene signatures for drug responses would be of broad interest in the field.

Major concerns,

1. It is not clearly stated how DEG was defined and called. What DEG method was used? What is the cutoff value for gene expression fold changes? There are no total DEG lists provided for both paired experimental conditions. These are important data source for GSEA analysis.

Minor concerns:

1. The sources, catalogue numbers, or any citations for Ion AmpliseqTM and Qlucore Omics Explorer 3.6 software are not mentioned.

2. “IL6-JAK-STAT3 signaling” is mentioned the first time here. Where is the support data in the results?

3. Figure 1E, F, G with blur labeling

4. Figure 2A, D scale bar labeling not clear; 2E, labeling too small and blur

5. Figure 3A, B, E, D, labeling blur

Author Response

Dear Reviewer,

Thank you for very valuable comments regarding the improvement of our manuscript ID Cancers-1778062 entitled " Proliferation and immune response gene signatures associated with clinical outcome to immunotherapy and targeted therapy in metastatic cutaneous malignant melanoma". We have taken your suggestions into consideration while revising the manuscript. Please find our response to the comments below. Changes are highlighted in yellow in the manuscript.

The paper entitled “Proliferation and immune response gene signatures associated with clinical outcome to immunotherapy and targeted therapy in metastatic cutaneous malignant melanoma” by Svedman et al. is an interesting paper. The authors used targeted sequencing of advanced melanoma biopsy tissues before targeted therapy or immunotherapy to identify gene signatures responsive to drug treatment. Although the sample size is limited, the discovered gene signatures for drug responses would be of broad interest in the field.

Major concerns,

  1. It is not clearly stated how DEG was defined and called. What DEG method was used?

Response: Standard score (z-score) normalization calculation was performed. This has now been added in section 2.3. “Z-score” has also been added to the scale bars in the figures.

What is the cutoff value for gene expression fold changes?

Response: A fixed fold-change of 1 was used for the two group comparison. This has now been added in section 2.3.

There are no total DEG lists provided for both paired experimental conditions. These are important data source for GSEA analysis.

Response: We have provided total DGE lists for both treatments, DC vs NR, PFS TT (linear regression) and PFS ICI (rank regression) in the Supplementary file, Supplementary Tables S15-S20. We apologize, but we forgot to add linear and rank regression analyses in section 2.3, which has now been added.

Minor concerns:

  1. The sources, catalogue numbers, or any citations for Ion AmpliseqTM and Qlucore Omics Explorer 3.6 software are not mentioned.

Response: The sources have now been added to section 2.3. Thermo Fisher Scientific, Waltham, MA USA for Ion AmpliseqTM and Qlucore, Lund, Sweden for Qlucore Omics Explorer 3.6 software.

  1. “IL6-JAK-STAT3 signaling” is mentioned the first time here. Where is the support data in the results?

Response: It is shown in Figure 1G and suppl Table S7. We have also now added “and IL6-JAK-STAT3 signaling with FDR < 10%” in the result text.

  1. Figure 1E, F, G with blur labeling

Response: We have improved the labeling.

  1. Figure 2A, D scale bar labeling not clear; 2E, labeling too small and blur

Response: We have made the scale bar labeling/labeling clearer and increased the labeling.

  1. Figure 3A, B, E, D, labeling blur

Response: We have made the labeling more clear.

Reviewer 2 Report

Svedman et al. have studied gene expression profiles related to treatment outcomes of BRAF+MEK inhibitors and immune checkpoint inhibitors in cutaneous melanoma. The aim of the study is important. Predictive biomarkers (other than BRAF-mutation) are needed in addition to clinical predictive factors (such as high LDH, tumor burden, performance status)  to guide treatment decisions in cutaneous melanoma. The manuscript is well-written and figures are good. The most obvious limitation is the small number of patients and the findings of this study require further validation in larger patient cohorts. However, the authors present novel findings and their methods are valid. Therefore, I could recommend this manuscript to be published in Cancers.

Author Response

Thanks for reviewing our manuscript and for positive comments.